# Peptides Derived from a Plant Protease Inhibitor of the Coagulation Contact System Decrease Arterial Thrombus Formation in a Murine Model, without Impairing Hemostatic Parameters

**DOI:** 10.3390/jcm12051810

**Published:** 2023-02-23

**Authors:** Daniel Alexandre De Souza, Bruno Ramos Salu, Ruben Siedlarczyk Nogueira, José Carlos Sá de Carvalho Neto, Francisco Humberto de Abreu Maffei, Maria Luiza Vilela Oliva

**Affiliations:** 1Laboratório de Química e Função de Proteínas, Departamento de Bioquímica, Universidade Federal de São Paulo, São Paulo 04044-020, SP, Brazil; 2Departamento de Cirurgia e Ortopedia, Universidade Estadual Paulista, Botucatu 18618-687, SP, Brazil

**Keywords:** coagulation hemostasis, kallikrein, peptides, protease inhibitors, thrombosis

## Abstract

Several plant protein inhibitors with anticoagulant properties have been studied and characterized, including the *Delonix regia* trypsin inhibitor (DrTI). This protein inhibits serine proteases (trypsin) and enzymes directly involved in coagulation, such as plasma kallikrein, factor XIIa, and factor XIa. In this study, we evaluated the effects of two new synthetic peptides derived from the primary sequence of DrTI in coagulation and thrombosis models to understand the mechanisms involved in the pathophysiology of thrombus formation as well as in the development of new antithrombotic therapies. Both peptides acted on in vitro hemostasis-related parameters, showing promising results, prolonging the Partially Activated Thromboplastin Time (aPTT) and inhibited platelet aggregation induced by adenosine diphosphate (ADP) and arachidonic acid. In murine models, for arterial thrombosis induced by photochemical injury, and platelet-endothelial interactions monitored by intravital microscopy, both peptides at doses of 0.5 mg/kg significantly extended the time of artery occlusion and modified the platelet adhesion and aggregation pattern with no changes in bleeding time, demonstrating the high biotechnological potential of both molecules.

## 1. Introduction

According to the World Health Organization, atherothrombotic diseases—atherosclerosis, thrombosis, and heart attack, among others—represent one of the greatest health challenges, accounting for more than 25% of deaths worldwide, with 80% of deaths occurring in underdeveloped and developing countries [1]. Atherothrombotic diseases are consequences of imbalances in hemostasis, which is defined as a set of complex biochemical processes that maintain the blood fluid and are contained in the vessels. Different systems involving several classes of proteins and cells are interrelated and regulated to maintain hemostasis in a perfectly coordinated biological dance [2].

Antithrombotic therapies are classified as either anticoagulants or antiplatelet agents depending on their target. Even before the introduction of the chemical principles of clinical therapeutics and random trials, anticoagulants, such as heparin and warfarin, and antiplatelet drugs, such as acetylsalicylic acid (ASA) and dicoumarol, were already utilized as antithrombotic therapies, even though the mechanism of action has not been elucidated [3]. Nowadays, anticoagulants can be classified as heparins, vitamin-K antagonists, and direct oral anticoagulants (DOACs) and are widely used worldwide. Antiplatelets can be classified as cyclooxygenase-1 (COX-1) inhibitors (Aspirin^®^), glycoprotein IIb/IIIa inhibitors (Abciximab^®^), protease-activated receptor antagonists, and ADP receptor antagonists [4]. The biggest challenge and limiting factor for most, if not all, antithrombotic drugs currently used in the clinic is the increased major bleeding risk [5].

Protease inhibitors from plants have been widely studied and differ in potency and specificity. Kunitz-type inhibitors have been found in several plants and have been characterized for decades for their ability to act as antimicrobial, antitumor, and anticoagulant agents [6,7,8,9,10,11]. The *Delonix regia* trypsin inhibitor (DrTI) is a Kunitz-type inhibitor with a molecular mass of approximately 20 kDa and two loops formed by four cysteine residues. Unlike other Kunitz inhibitors, which contain either arginine or lysine, glutamic acid is found at the reactive site position (residue 63, Soybean Trypsin Inhibitor) [12]. The structure of DrTI consists of 12 antiparallel β-sheets connected by long loops that form six β-clamps, each formed by two strands. Three of the β-clamps form a barrel structure, whereas the others form a triangular cover on the barrel [13]. Regarding its biological activity, DrTI inhibited trypsin (Ki_app_ = 2.2 × 10^−8^ M), human plasma kallikrein (PKa; Ki_app_ = 5.25 × 10^−9^ M), factor XIIa (FXIIa; Ki_app_ = 3.1 × 10^−7^ M), and factor XIa (FXIa; Ki_app_ = 1.3 × 10^−6^ M), as well as ex vivo ADP-mediated platelet aggregation [10]. The DrTI also prolonged the time required for occlusion of the left carotid artery in a model of arterial thrombosis induced by a photochemical injury without increasing the bleeding time [10].

In this study, we describe two novel peptides whose design is based on the structure of the protease inhibitor, DrTI. These peptides inhibit coagulation proteases, affect platelet activity both in vitro and in vivo, and prolong the time for arterial occlusion in murine models of thrombosis, doing all of that without impairing hemostatic parameters, such as bleeding time and ex vivo coagulation tests, and showing great biotechnological potential.

## 2. Materials and Methods

### 2.1. Reagents

Bovine trypsin was purchased from Roche (Mannheim, Germany) and human plasma kallikrein (PKa) was purified in the Laboratory of Chemistry and Protein Function using the protocol established by Prof. Dr. Maria Luiza Vilela Oliva and colleagues in 1982 [14].

The substrates Boc-Glu(Obzl)-Ala-Arg-AMC.HCl, Bz-DL-Arg-pNA.HCl and H-D-Pro-Phe-Arg-pNa were purchased from Bachem (Bubendorf, Switzerland). Thromborel S, thrombin, and Dade actin-activated cephaloplastin were obtained from Dade Behring (Marburg, Germany), and heparin was obtained from Roche (liquenime^®^). Rose Bengal (3,4,5,6 tetrachloro-2,4,5,7-tetraiodofluorescein) and rhodamine 6G were purchased from Sigma Aldrich (Saint Louis, MO, USA). ADP, arachidonic acid, and collagen I were obtained from Chrono-log (Pennsylvania, CA, USA).

### 2.2. Plant Material

The protein inhibitor DrTI was isolated and purified from *Delonix regia* seeds, as previously described [12]. The seeds were collected in the city of Manaus, and the exsiccate was deposited in the herbarium of the State University of Bahia (UEDB) (HUESB 12431). The studies were carried out under license N° 02/2014, process 020 00.003472/2005-62 of the Ministério do Meio Ambiente, Coordenação Geral de Autorização de uso da Flora e Floresta, SCEN.

### 2.3. Peptides

Two peptides derived from the primary structure of the natural inhibitor DrTI (*Delonix regia* trypsin inhibitor), and synthesized by Watsonbio (Houston, TX, USA), were used and their homogeneity, purity, and concentration were evaluated using reversed-phase high-performance liquid chromatography. Both peptides are mirror images, and the pDrTI-RI peptide sequence is the inverse of the pDrTI sequence (Table 1). However, pDrTI-RI was synthesized with D-amino acids, making pDrTI-RI and pDrTI enantiomers.

### 2.4. Animals

To evaluate the effect of the purified plant inhibitor and peptides derived from its sequence, adult male C57BL/6J mice weighing 20–25 g were obtained from the Animal Facility of the Universidade Federal de São Paulo, Center of Development of Experimental Models for Biology and Medicine (CEDEME). The animals were maintained at 23 °C on a 12 h light/dark cycle with water and food ad libitum. The project was approved by the Ethics Committee on the Use of Animals of UNIFESP (CEUA UNIFESP) under number 7922240819. The studies were carried out following international principles of animal use, such as those established in the Manual on Care and Use of Laboratory Animals (ILAR, 1996) and Federal Law 6638 of 8 May 1979. The animals were treated in accordance with the ethical principles of animal experimentation developed by the National Council for the Control of Animal Experiments (CONCEA) and established by the Canadian Council on Animal Care and the Guide for the Care in Use of Animals from the NIH. After experimentation, the animals were euthanized and disposed of, as determined by the Committee on Ethics in the Use of Animals (CEUA).

### 2.5. Plasma and Platelets

Blood from healthy volunteers was collected by venipuncture of the basilic vein in a syringe containing 1/10 volume of 3.8% sodium citrate solution. The collection was performed slowly to avoid blood hemolysis. Blood was centrifuged at 3000× *g* for 10 min at 25 °C to obtain platelet-rich plasma (PRP), which was then centrifuged again at 5000× *g* for 10 min at 25 °C to obtain plasma poor in platelets (PPP). Plasma was aliquoted and stored at −80 °C for further testing. Platelet bags, kindly provided by the Beneficent Association of Blood Collection (COLSAN, São Paulo), were used for the platelet aggregation assays. Basal platelet activity was evaluated before all experiments.

### 2.6. Inhibitory Activity

The inhibitory activity of the peptides and the purified protein was measured using clotting enzymes and their chromogenic substrates in 96-well plates at 37 °C. Briefly, enzymes were incubated with peptides or saline for 10 min. The reactions were initiated by the addition of the substrate, and the reaction was continuously monitored using a plate spectrophotometer.

For trypsin inhibition (37 nM), the substrate Bz-DL-Arg-pNA.HCl (1 mM) was used in 0.05 M Tris-HCl buffer (pH 8.0) containing 0.2% CaCl_2_. Inhibitors were incubated for 10 min with the enzyme at a concentration of 25 µM. The substrate was then added, and the reaction was monitored for 30 min with readings every 5 min. Substrate hydrolysis was verified at 405 nm using a microplate reader (Spectra CountTM). For confirmation, inhibition was performed using endpoint readings with increasing concentrations of the inhibitors under study.

For PKa inhibition (14.7 nM), the substrate HD-Pro Phe-Arg-pNa (2.5 mM) was used in 0.05 M Tris-HCl buffer (pH 7.4) containing 12 mM NaCl and 1 mg/mL bovine serum albumin. The enzyme was incubated with the inhibitors at a concentration of 25 µM for 10 min and then the substrate was added. Reactions were performed for 30 min, with readings every 5 min. Absorbance was read at 405 nm using a microplate reader (Spectra CountTM). Inhibitors were also evaluated at increasing concentrations in assays with endpoint readings.

### 2.7. Coagulation Assays

For the in vitro coagulation assays, the pool of blood plasma was used as described previously, measurements were performed in triplicate, and the results were expressed relative to the control. Assays were performed using a semi-automatic coagulometer (BTF II, Dade Behring). For the ex vivo assays, the compounds were administered intraperitoneally. After 10 min, the animals were anesthetized with ketamine (100 mg/kg) and xylazine (10 mg/kg). Once anesthesia was confirmed, blood was collected by cardiac puncture with a 1 mL syringe containing 3.8% sodium citrate in a 1:9 ratio, and the animals were euthanized. The PPP was obtained as described in Section 2.5.

Activated partial thromboplastin time (aPTT) was determined using a semiautomatic coagulometer (BFT II, Dade Behring) using activated cephalin from rabbit brain containing celite as an activator. PPP (50 µL) was pre-incubated at 37 °C for 120 s with 50 µL of cephalin and 50 µL of different concentrations of the inhibitors. After pre-incubation, 50 µL 0.025 M CaCl_2_ was added at 37 °C. The reaction was timed and the results were expressed in seconds, with a maximum time of 300 s.

### 2.8. Platelet Aggregation

For platelet aggregation, experiments were performed using a Chrono-log 700 aggregometer. Once the activity was confirmed, the platelets were diluted in HEPES-Tyrode buffer (NaCl 137 mM, KCl 2.90 mM, CaCl_2_ 1.00 mM, MgCl_2_ 1.00 mM, NaH_2_PO_4_ 12 mM, HEPES 5 mM, and Glucose 5.56 mM, pH 7.4), and its concentration was adjusted to approximately 3.5 × 10^5^ platelets/mL. Different concentrations of the inhibitors were incubated with PRP in 500 µL cuvettes under constant agitation at 37 °C for 10 min. At the end of the incubation period, an agonist (2 mM ADP, 1 mM arachidonic acid, or 1 µg/mL collagen I) was added, and the reaction was monitored by measuring the transmittance at 595 nm.

### 2.9. Photochemical-Induced Arterial Thrombosis Model

Arterial thrombosis was induced according to the procedure described by Eitzman et al. with modifications [15]. After anesthesia, the substances were intravenously administered into the left orbital plexus. Five minutes after administration, the left common carotid artery was dissected through a longitudinal incision near the tracheal region of the animal. After dissection, an ultrasound probe (Transonic Systems) was placed in the artery to monitor blood flow, and Rose Bengal dye (Sigma-Aldrich, Saint Louis, MO, USA) was administered through the right retro-orbital plexus at 50 mg/kg, solubilized in 0.15 M NaCl, and protected from light. Immediately after dye administration, a laser with a wavelength of 542 nm (5 mm in diameter) was focused on the artery 6 cm away. It remained focused until the arterial blood flow reached 0.05 mL/min or lower, demonstrating complete obstruction of the artery. At the end of the experiment, the animals were euthanized with an excessive dose of ketamine and xylazine, and euthanasia was confirmed by cervical dislocation.

### 2.10. Platelet Behavior in Intravital Microscopy

After anesthesia, the carotid arteries of the animals were isolated and dissected as described above. After dissection, the substances under study were administered intravenously through the retro-orbital plexus, and 5 min later, rhodamine 6G (Sigma-Aldrich, Saint Louis, MO, USA) (1 mg/kg) was also administered via retro-orbital injection in the opposite optic cavity to label mitochondria from platelets and leukocytes. For the aspirin control, aspirin diluted in acidic medium was administered by gavage 30 min before the dissection procedure. The arterial injury was induced after 5 min by applying, for 2 min, a piece of filter paper (2 × 1 mm) saturated with 15% FeCl_3_ (Sigma-Aldrich, Saint Louis, MO, USA). The injured artery was then monitored for 12 min (defined as the maximum fluorescence time) using an intravital fluorescence microscope (Carl Zeiss Imager.A2) coupled with an Axiocam HSm camera (Carl Zeiss, Jena, Germany). After obtaining images, the animals were euthanized.

### 2.11. Bleeding Time

The procedure described by Perzborn et al. [16], with some modifications, was used. Animals were anesthetized with ketamine (100 mg/kg) and xylazine (10 mg/kg). Ten min after the administration of the peptide inhibitors under study or saline solution (NaCl 0.9%), a 2 mm longitudinal incision was made at the end of the tail of the mice. The cut tail was immersed in saline solution (NaCl 0.9%) at a temperature of 37 °C. The time when the bleeding stopped for >30 s was measured, with a maximum observation time of 10 min.

### 2.12. Statistical Analysis

As the original data were not normally distributed, all analyses were based on nonparametric methods. Statistically significant differences were determined using Kruskal-Wallis tests complemented by Dunn’s multiple comparisons with significance set at 5% (*p* < 0.05). Data are reported as mean ± standard error. Statistical summary data were graphically presented using box plots. All analyses were performed using SPSS version 20 (SPSS Inc., Chicago, IL, USA).

## 3. Results

### 3.1. Inhibitory Activity

Blood plasma contains several zymogens that are precursors of serine proteases with trypsin-like activity [17]; thus, bovine trypsin inhibition was used as the initial step in characterizing coagulation protease inhibitors. Therefore, the inhibitory activities of purified DrTI and peptides derived from its primary structure were determined by the inhibition of bovine trypsin. Inhibition of bovine trypsin with DrTI has already been characterized, with Ki_app_ = 21.9 nM [12]. Concerning the inhibitory properties of the peptides, when they were used at the same concentration as DrTI (Figure 1A), no inhibition of bovine trypsin was observed. This result was expected because the peptides were not designed based on the reactive site of the protein [13]. To ensure that the absence of inhibitory effects was related to the peptides and not to their concentration, endpoint inhibitory assays were performed using higher peptide concentrations (Figure 1B). At 680 µM, none of the peptides inhibited bovine trypsin activity.

Pka is produced in the liver, is involved in coagulation, fibrinolysis, blood pressure regulation, and inflammatory reactions, and is one of the main enzymes in the kallikrein-kinin system. The DrTI protein is a potent inhibitor of Pka, with Ki_app_ = 5.25 nM with 1:1 stoichiometry [10]. Similar to trypsin inhibition, inhibition was performed with the peptides at the same concentration as the protein (25 µM) (Figure 1C). At concentrations equal to those of the protein, none of the peptides inhibited the catalytic activity of Pka on the chromogenic substrate. To ensure that the absence of inhibitory activity was not concentration-related, inhibitory endpoint experiments were performed using higher peptide concentrations. The pDrTI peptide was able to inhibit Pka at high concentrations (Figure 1D), whereas the pDrTI-RI peptide did not alter its activity, even at the highest concentration tested. Inhibition by pDrTI can be explained by the same tripeptide sequence found in the peptide and chromogenic substrate used, Pro-Phe-Arg, resulting in competitive inhibition. The inhibitor pDrTI-RI also contains the tripeptide sequence as it is an enantiomer; however, the sequence is inverted Arg-Pro-Phe, with D-amino acids, indicating stereospecificity in the catalytic activity of plasma kallikrein.

### 3.2. Coagulation Assays

In vitro aPTT assays were performed in human plasma obtained from a pool of five healthy individuals. The initial concentration of the peptides to be tested was defined as 0.55 mM, and for DrTI, 25 µM was used since at approximately 21 µM, the protein is able to prolong the aPTT two-fold [10].

At the concentration used, the pDrTI peptide prolonged clot formation in the aPTT test 1.5 folds the time required in comparison with non-treated control, whereas the positive controls prolonged it by approximately three times (DrTI), and heparin left the plasma incoagulable (Figure 2A). The results found for the pDrTI peptide corroborate the results found in the inhibitory tests, since the inhibition of PKa had little effect on the alteration of aPTT compared to the inhibition of proteins directly involved in the extrinsic pathway, such as FXIa. A dose–response relationship with aPTT prolongation was observed for the pDrTI-RI peptide (Figure 2B). At the highest concentration used, the prolongation was eight fold the control time, which was even greater than that observed for the protein. At a concentration of 0.27 µM, the peptide prolonged clot formation approximately at two times the time required.

The inhibitory activity of the pDrTI-RI peptide on intrinsic coagulation pathway enzymes and possibly FXIa was determined in the plasma inhibition assay. When ex vivo aPTT tests were performed, none of the peptides were able to change the clot formation time (Figure 2C) at the analyzed concentrations. We hypothesize that this is due to the fact that the concentration used for the ex vivo aPTT experiments is higher than the concentrations used for the other in vivo tests and is significantly lower than the concentrations used in the in vitro tests. There were no changes in aPTT in vitro at concentrations similar to those used in the in vivo assays.

### 3.3. Platelet Aggregation

As already described in the literature, DrTI can inhibit ADP-mediated platelet aggregation ex vivo in blood aggregometry [10]. The inhibitory effect on platelet aggregation was maintained when a PRP assay was performed. DrTI treatment exhibited a dose-response but required higher concentrations compared to concentrations used in inhibition assays. The peptides were tested for ADP-induced platelet aggregation in human PRP and the peptide pDrTI-RI showed better inhibition of platelet aggregation compared to the peptide pDrTI, and both showed higher inhibition values compared to the protein at the same concentrations (Figure 3A,B). It is important to note that the peptides had an effect at very low concentrations, in which coagulation protease inhibition tests showed no effect, indicating that the in vivo action of these peptides is related to changes in platelets and not to coagulation factors. Although it had a smaller effect, arachidonic acid induction achieved the same result as observed for induction by ADP (Figure 3C,D). Both peptides completely inhibited arachidonic acid-induced platelet aggregation at 200 µM. Interestingly, the peptides were not able to inhibit collagen-induced platelet aggregation (Figure 3E), suggesting that antiplatelet action occurs via mechanisms other than the collagen-mediated activation pathway.

### 3.4. Photochemical-Induced Arterial Thrombosis Model

The preferential effect of the peptides on platelets concerning clotting factors led us to choose an arterial thrombosis model as opposed to a venous thrombosis model. The thrombus in vessels with high shear stress, such as arteries, is mostly formed by platelets, with a lower concentration of clotting factors in relation to venous thrombus [18]. The model of arterial thrombosis induced by photochemical injury causes a lesion in the endothelium and, consequently, activates platelets and coagulation factors by exposing negatively charged surfaces on the subendothelium, which has already been characterized in this model, prolonging the time required for artery occlusion [10,19]. The effect of the peptides in this model was evaluated against published data for DrTI and heparin. Both peptides prolonged the time required for artery occlusion and the consequent stoppage of blood flow at concentrations of 100 µM, equivalent to a dosage of 0.5 mg/kg (Figure 4A). The pDrTI peptide’s time to occlusion was 88.6 ± 9 min and the pDrTI-RI peptide took 97.8 ± 22 min. The control group took 53 ± 7 min for artery occlusion and heparin took 68.5 ± 14 min. The two peptides showed a significant difference in relation to the negative control (NaCl 0.9%) and the positive control (heparin), with *p* < 0.05 in the *t*-test with Welch’s correlation. Heparin, usually used in clinics for anticoagulation, showed no significant difference compared with the negative control (Figure 4B).

These data demonstrate the potential of these peptides as antithrombotic drugs. The time required for occlusion of the carotid artery was approximately two-fold that of untreated mice, and these times were similar to those found for the native protein and other Kunitz-type inhibitors such as AsTI and BbKI [8,10].

### 3.5. Platelet Behavior in Intravital Microscopy

Considering the action of peptides on platelets and the existence of an increase in the time required for arterial occlusion, the effect of peptides on platelet behavior was investigated by intravital microscopy. Initially, the maximum fluorescence time in the control (12 min) and platelet behavior profile for DrTI and aspirin were defined (Figure 5). We observed that the protein did not allow the total adhesion of platelets to the vascular wall, with a maximum fluorescence at 6 min after the induction of the lesion by 15% ferric chloride. In the following minutes, a decrease in fluorescence was observed, indicating the de-adhesion of previously recruited platelets. As for the mice treated with aspirin, there was no difference in terms of the control, which can be explained by the fact that the method of induction by FeCl_3_ is very aggressive and aspirin inhibits COX-1 in a time-dependent manner [20]. Another factor to be considered is that although the COX-1/COX-2 proteins of rabbits and mice are similar to those of humans, the amino acids that constitute the reactive site are not necessarily conserved [21].

Both peptides demonstrated the same pattern of platelet behavior as that observed for the protein (Figure 6). Although the effects were visually smaller than those of the native protein, the behavior followed a similar pattern, with an increase in fluorescence at 9 min and a decrease at 12 min. The release of large platelet aggregates that had previously adhered to the lesion site was also observed.

### 3.6. Bleeding Time

Peptides have already been shown to prolong the occlusion time in arterial thrombosis, do not change ex vivo coagulation assays, and appear to act mainly on platelets and thrombus propagation, a combination of factors favorable to the absence of bleeding. With this in mind, bleeding time assays, or induced bleeding time, was performed in mice using these peptides (Figure 7).

While heparin made the mice uncoagulable, neither peptide significantly altered the time required to stop the bleeding. The pDrTI-RI peptide appeared to increase the coagulability of the animal in this model, which should be further investigated, while the pDrTI peptide maintained, on average, the same time required for the untreated animal’s blood to clot.

## 4. Discussion

Kunitz-type inhibitors, which have been characterized for decades, are present in several plants, have been widely studied, differ mostly in potency and specificity of inhibition, and can act as antimicrobial, antitumor, and anticoagulant agents in vivo [6,7,8,10,11].

Among these inhibitors, DrTI stands out for its ability to inhibit PKa and FXIa, which are involved in several biological pathways, such as the intrinsic coagulation pathway, complement system, renin-angiotensin system, and kallikrein-kinins system, as well as in fibrinolysis and inflammation [22]. In recent years, FXIa has been considered a potential target for antithrombotic drugs, as individuals with FXIa deficiency (hemophilia C) have a lower risk of bleeding than those with factor VIII deficiency (hemophilia A) or factor IXa deficiency (hemophilia B) [23].

DrTI has notable characteristics. In addition to inhibiting proteins that are at the center of attention in antithrombotic research, DrTI is capable of inhibiting platelet activity and acting as an arterial antithrombotic without altering bleeding time, which is considered the greatest limiting factor of these therapies. However, owing to the size of proteins, the importance of the three-dimensional protein structure to function, which is maintained by weak non-covalent forces, and the cost of extraction and purification in large quantities makes the use of proteins for basic characterization and clinical use an expensive process. Other challenges faced in the search for proteins as biopharmaceuticals are their multifaceted mechanisms of action as well as the potentially high number of degradation products, which can act in several parallel biochemical processes [24].

Consequently, the Laboratory of Chemistry and Protein Function group at UNIFESP, led by Prof. Dr. Maria Luiza Vilela Oliva, explored protein fragments capable of maintaining antithrombotic action without increasing bleeding, avoiding most of the limitations found in the use of proteins, which is the case for the peptides studied in this work.

In recent years, owing to their size and the balance between structural rigidity and versatility, peptides have become promising candidates as inhibitors with satisfactory affinity and specificity and fewer side effects and toxicity. Deciphering and characterizing the mechanisms of interaction between peptides and proteins is imperative for the development of strategies to interfere with the action of endogenous proteins or to increase the affinity and specificity of existing strategies. Several methods have been used to design peptides using biological approaches, such as phage display, or in silico approaches, such as molecular docking [25].

Usually, for the design of peptides with the same activity as the natural inhibitor, the structure of the binding site of the protein is used as a basis; however, for the peptides used in this study, after analyzing the primary structure of DrTI, we found a tetra-peptide sequence—Ser–Pro-Phe-Arg—similar to that found in the vasoactive peptide bradykinin [26]. This led to the question of why an inhibitor of plant origin has the same sequence as an endogenous peptide. With this in mind, two peptides containing the tetra-peptide sequence and six amino acid residues found downstream in the primary sequence of the inhibitor were synthesized.

Both peptides are mirror images, and the pDrTI-RI peptide sequence was the inverse of the pDrTI sequence. However, pDrTI-RI was synthesized with D-amino acids, making pDrTI-RI and pDrTI enantiomers. The use of d-amino acids is a peptide modification strategy aimed at greater stability, since these amino acids rarely serve as substrates for endogenous proteins, making them more resistant to proteolysis and increasing their half-life [27].

In this way, the first phase of the work to characterize the action of these peptides consisted of protease inhibition assays, in which the native protein has already been characterized by its inhibitory activity: bovine trypsin, PKa, and FXIa. Our results demonstrated that the two peptides were unable to inhibit these three enzymes at concentrations similar to those used by the protein; however, at high concentrations, the peptide pDrTI inhibited PKa activity and pDrTI-RI inhibited FXIa activity. Some PKa inhibitors are also capable of inhibiting FXIa activity, as the proteins are 58% similar [28]. However, in the case of our peptides, even as enantiomers with essentially the same sequence, each peptide was able to act on one of these enzymes without interfering with the other, indicating stereospecificity in the action of the peptides on these proteins, showing high specificity of action.

The aPTT coagulation assay is widely used clinically and in basic research to assess coagulation changes. aPTT is a screening test sensitive to changes in the intrinsic coagulation pathway and can indirectly measure the activities of factors II, V, VIII, IX, XI, XII, kallikrein, high molecular weight kininogen, and reduced fibrinogen concentration [29]. The action of the peptides on this parameter was consistent with their respective inhibitions, since the inhibition of kallikrein by pDrTI has an indirect effect on the intrinsic coagulation pathway, while the inhibition of FXIa has a more pronounced effect. None of the cases showed ex vivo changes, suggesting that the in vivo action occurred only on platelets.

In the case of platelet aggregation, the two peptides were able to inhibit the aggregation induced by ADP and arachidonic acid at concentrations similar to those of the native protein, with greater effectiveness compared to the native protein. The action of these peptides on platelet aggregation may occur in a pathway common to these two agonists after platelet activation, such as the release of α-granules or thrombin generation, or acting separately on the two pathways, which may be investigated by measuring the levels of thromboxane B2 generated [30].

In view of the protease inhibition and platelet aggregation results, the concentration for thrombosis models was set at 100 µM, equivalent to a dosage of 0.5 mg/kg. According to the data obtained in this work, at this concentration, the peptides are capable of inhibiting platelet aggregation but do not act on coagulation proteases. The pDrTI peptide took 88.6 ± 9 min for complete occlusion and the pDrTI-RI peptide took 97.8 ± 22 min, much higher than the control group, whose time to total artery occlusion was 53.0 ± 7 min, while heparin, used as a positive control, was 68.5 ± 14 min. On average, the peptides increased the time required for artery occlusion by two-fold, comparable to other protein inhibitors previously described.

Arterial thrombi are mostly formed by platelets, and a decrease in platelet aggregation is a good alternative to alleviate the pathological process, as has already been demonstrated by the use of drugs such as aspirin and clopidogrel [4]. In an arterial model induced by ferric chloride, platelets from mice treated with peptides appeared to adhere to the lesion site but failed to overcome shear stress and remain aggregated. These data suggested that the peptides do not act directly on the adhesion of platelets to the lesion site, mainly due to platelet aggregation and consequently decrease thrombus propagation. Therefore, these peptides could maintain the balance between stopping bleeding upon injury, maintaining restoration of the injured vessel, and preventing pathological thrombus formation. Interestingly, these results are comparable to those found in bradykinin B2 receptor knockout mice [31]. In fact, owing to the structural similarity of the peptides to bradykinin, some of the effects found may be related to the interaction with the bradykinin B2 receptor. Notably, a stable metabolite of bradykinin inhibits thrombin-induced platelet aggregation. Furthermore, nitric oxide released by the interaction of bradykinin with the bradykinin B2 receptor is a potent inhibitor of platelet adhesion and aggregation [32]. This mechanism also leads to vasodilatation, which explains abnormal bleeding in heparin-treated individuals [33]. The effects of peptides on platelet adhesion should be evaluated in future studies.

The biggest challenge and limiting factor in antithrombotic therapies is bleeding. Currently, most, if not all, antithrombotic drugs used in the clinic have bleeding as one of their side effects. The search for compounds capable of protecting against thrombosis without increasing the risk of bleeding is a holy grail of basic research on this subject [34]. The dataset presented in this work demonstrates the potential of the peptides studied, indicating that their effect in vivo occurs in platelets and can increase the time required for arterial thrombus formation without altering coagulation tests and without increasing bleeding time, which are desirable characteristics for antithrombotic drugs. However, further studies must be carried out to definitively characterize its action, toxicity, and pre-formulation, aiming at the establishment of a bioproduct.

## 5. Conclusions

In conclusion, the DrTI-derived mimetic peptides studied in this work maintained the main characteristics of the protein, its action on platelets, and the increased time required for arterial occlusion by thrombus formation with no prolongation of bleeding time. Although the mechanism of action of these peptides has not been characterized, their potential as possible antiplatelet drugs or as a biotechnological tools for the study of platelets is remarkable.

## Figures and Tables

**Figure 1 jcm-12-01810-f001:**
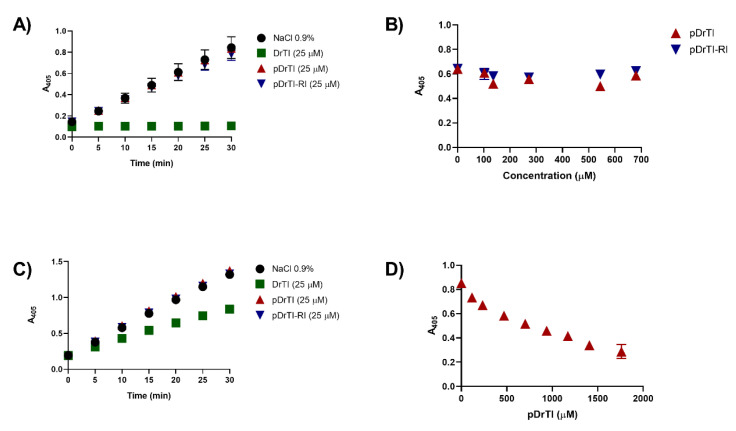
Inhibitory activity—Bovine trypsin (37 nM) activity was performed with the synthetic substrate BAPNA (Bz-Arg-pNA). Substrate hydrolysis was measured through p-nitroaniline release by reading on a plate spectrophotometer at a wavelength of 405 nm every 5 min (**A**) or the end of the 30 min (**B**) with different concentrations of the peptides under study. PKa (14.7 nM) activity was performed with the synthetic substrate HD-Pro-Phe-Arg-pNa. Substrate hydrolysis was measured through p-nitroaniline release by reading on a plate spectrophotometer at a wavelength of 405 nm every 5 min (**C**) or the end of the 30 min (**D**) with different concentrations of the peptides under study. Assays were done in triplicate.

**Figure 2 jcm-12-01810-f002:**
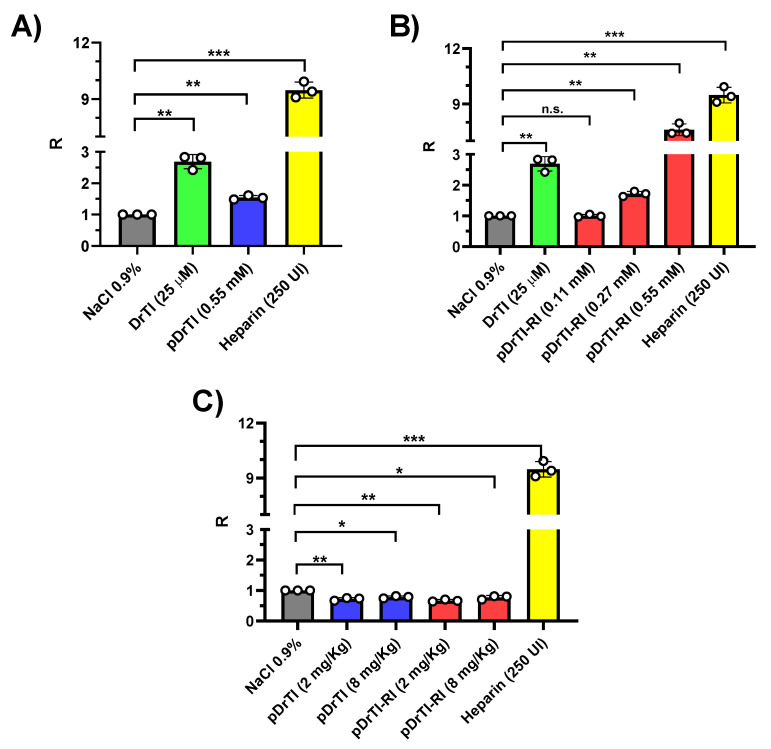
aPTT tests—Coagulation assay (aPTT) performed in vitro with human plasma in a Dade Behring semiautomatic coagulometer. PPP was incubated with pDrTI (**A**) or pDrTI-RI (**B**) in the presence of cephalin. After incubation, 0.02% CaCl_2_ was added and the time for fibrin clot formation was recorded. (**C**) Coagulation assay ex vivo. Animals were pre-treated by retro-orbital injection with the substances 30 min prior to euthanasia. Whole blood was collected by cardiac puncture with 3.8% sodium citrate, 1:9, and centrifuged to obtain PPP. Then, a coagulation assay was performed in a Dade Behring semiautomatic coagulometer. R = Ratio between the clotting time of the compounds under study and the control. Assays were performed in triplicate. Statistical analysis was made using *t*-test with Welch’s correction using GraphPad Prism 8. n.s.: non-significant; * *p* < 0.05; ** *p* < 0.01; *** *p* < 0.001.

**Figure 3 jcm-12-01810-f003:**
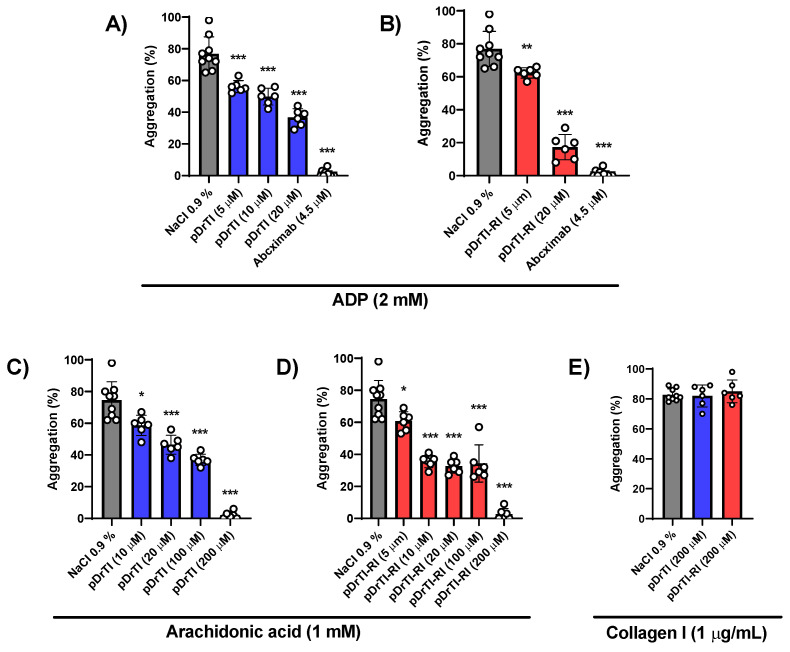
Platelet Aggregation—PRP was obtained from healthy individuals. Different concentrations of the peptides were added to the PRP and incubated at 37 °C for 5 min. After the incubation time, the agonist (ADP, arachidonic acid, or collagen I) was added and the reaction was monitored in an automatic aggregometer Chrono-log 700 through transmittance/impedance. Assays were performed in triplicate. Results are shown as percentage aggregation, ADP induced aggregation was inhibited by pDrTI (**A**) and pDrTI-RI (**B**); Arachidonic acid induced aggregation was inhibited by pDrTI (**C**) and pDrTI-RI (**D**) and collagen I induced aggregation was not inhibited (**E**). Statistical analysis was made using one-way ANOVA with Welch’s correction using GraphPad Prism 8. * *p* < 0.05; ** *p* < 0.01; *** *p* < 0.001.

**Figure 4 jcm-12-01810-f004:**
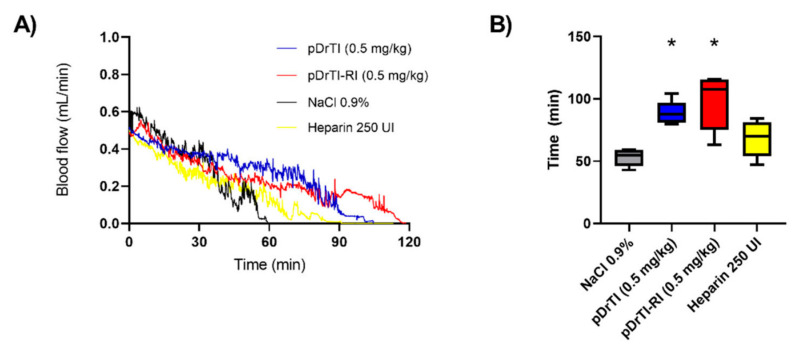
Arterial Thrombosis—The animals were pre-treated with pDrTI, pDrTI-RI, heparin, or NaCl by retro-orbital intravenous injection. The photosensitive dye Rose Bengal was administered and the lesion was induced through a laser incision λ = 542 nm (5 mm diameter) 6 cm away from the artery, and the flow was monitored through a Doppler probe until it was less than or equal to 0.05 mL/min. Blood flow is shown in (**A**) and occlusion time in (**B**). * *p* < 0.05. N = 5 animals.

**Figure 5 jcm-12-01810-f005:**
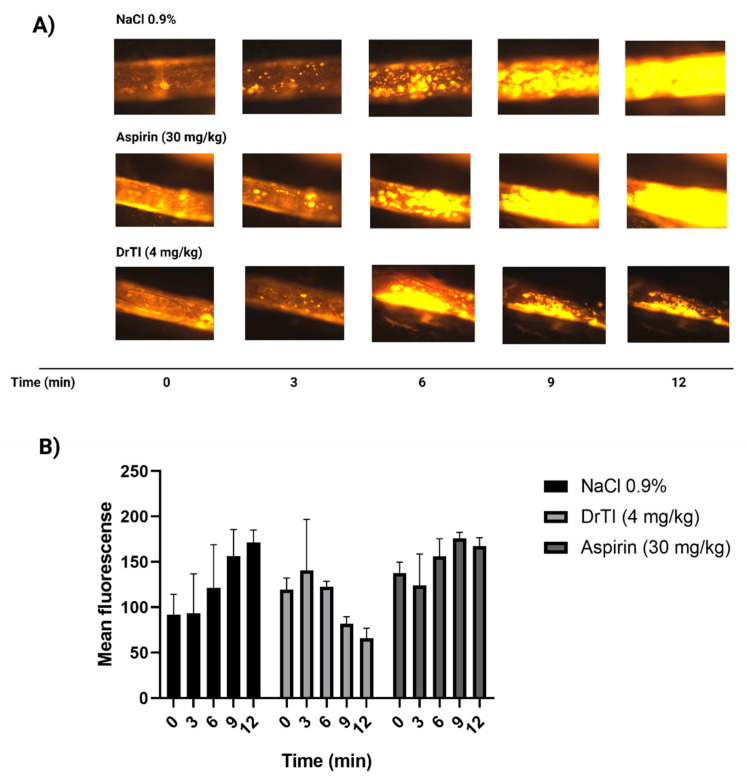
Platelet behavior after treatment with DrTI and Aspirin—(**A**) DrTI (4 mg/kg), Aspirin (30 mg/kg), and NaCl (0.9%) were administered by retro-orbital injection and aspirin by gavage 30 min before dissection. Rhodamine 6G was administered by retro-orbital injection to label mitochondria in platelets and leukocytes. Arterial injury was induced by the application of filter paper saturated with 15% FeCl_3_. The injured artery was monitored for 12 min with an image capture every 3 min using an intravital fluorescence microscope. (**B**) Mean fluorescence was quantified using the software ImageJ version 1.46r. N = 3 animals.

**Figure 6 jcm-12-01810-f006:**
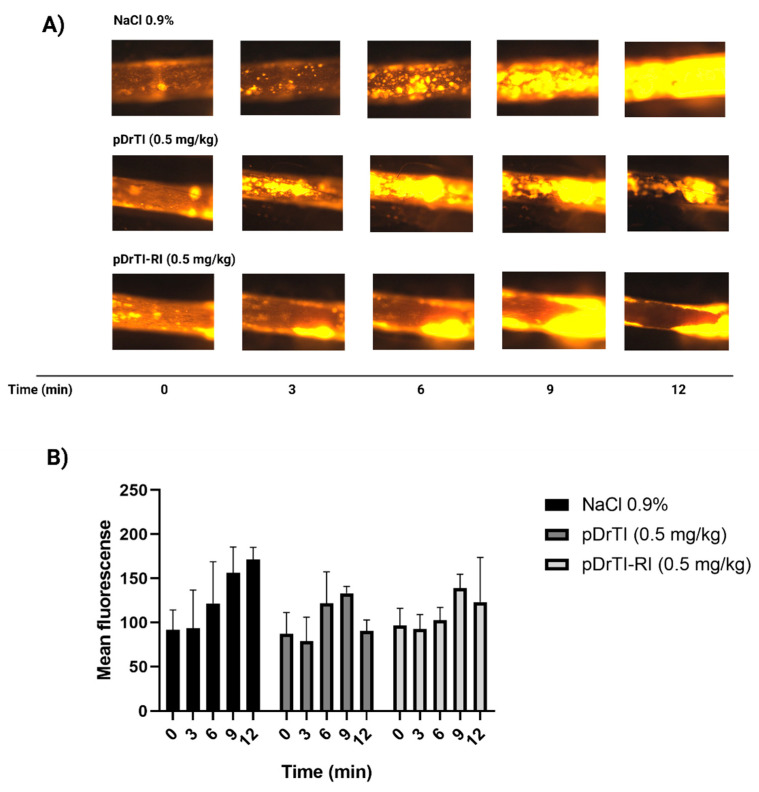
Platelet behavior after treatment with peptides—(**A**) pDrTI (0.5 mg/kg) or pDrTI-RI (0.5 mg/kg) were administered by retro-orbital injection before dissection. Rhodamine 6G was administered by retro-orbital injection for mitochondrial platelet and leukocyte labeling. The arterial injury was induced by the application of filter paper saturated with 15% FeCl_3_. The injured artery was monitored for 12 min with an image capture every 3 min using an intravital fluorescence microscope. (**B**) Mean fluorescence was quantified using the software ImageJ version 1.46r. N = 3 animals.

**Figure 7 jcm-12-01810-f007:**
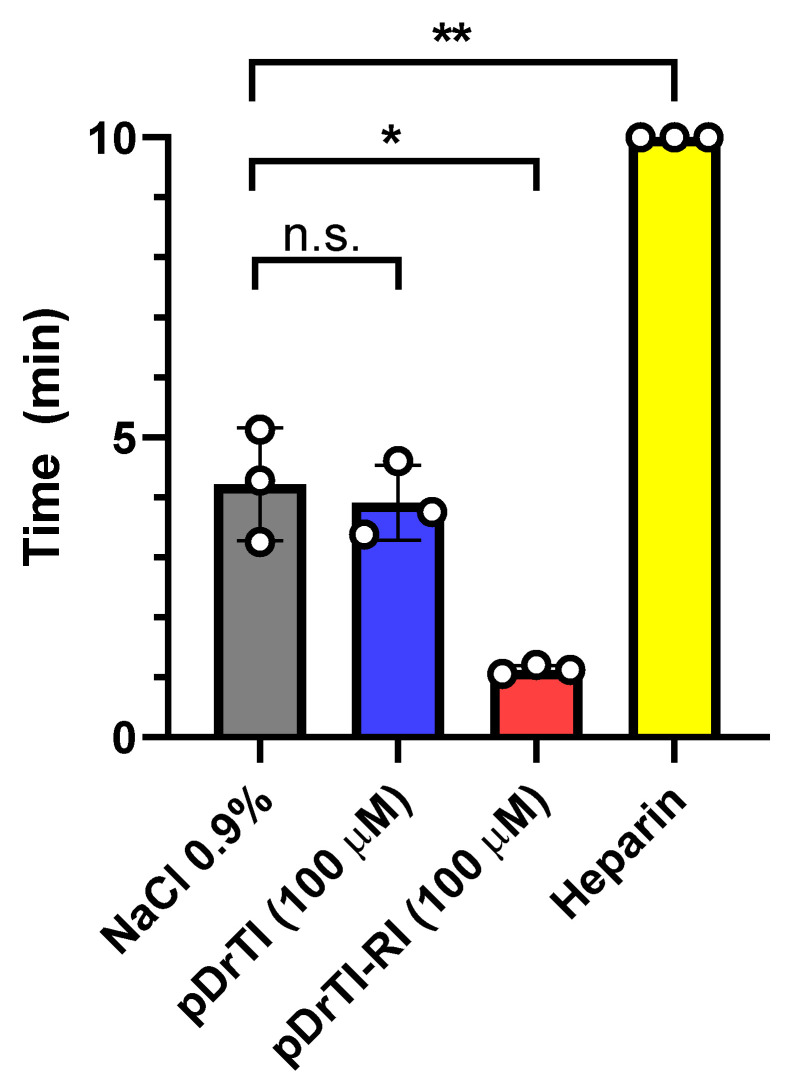
Tail vein bleeding time—Mice were treated with pDrTI, pDrTI-RI, or heparin. After 10 min, a 2 mm longitudinal incision was made at the end of the mouse’s tail. The sectioned tail was immersed in a saline solution (NaCl 0.9%) at 37 °C and the time required for bleeding cessation was recorded for up to 10 min. N = 3 animals. Statistical analysis was made using *t*-test with Welch’s correction using GraphPad Prism 8. n.s.: non-significant; * *p* < 0.05; ** *p* < 0.01.

**Table 1 jcm-12-01810-t001:** Peptides sequence.

Peptide	Sequence
pDrTI	Ac-S-P-F-R-V-V-F-V-K-P-NH_2_
pDrTI-RI	Ac-P *-K *-V *-F *-V *-V *-R *-F *-P *-S *-NH_2_

* d-amino acids.

## Data Availability

Datasets used and/or analyzed during the current study are available from the corresponding author upon request.

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
