# Peer review of "Peptides Derived from a Plant Protease Inhibitor of the Coagulation Contact System Decrease Arterial Thrombus Formation in a Murine Model, without Impairing Hemostatic Parameters"

_jcm, 2023, doi:10.3390/jcm12051810_

Round 1

Reviewer 1 Report

In this study the authors have investigated the effects of two plant-derived peptides on coagulation hemostasis. This work builds on an extensive research program of the lead author on the mechanisms of proteases in coagulation and thrombosis. The two enantiomeric peptides were designed based on the Delonix regia trypsin inhibitor (DrTI) and bradykinin consensus sequence Ser-Pro-Phe-Arg. The key findings described in this manuscript are as follows: at the concentrations used for the protein DrTI, the peptides pDrTI and pDrTI-RI were unable to inhibit the coagulation enzymes trypsin, factor XIa and plasma kallikrein; the peptides inhibited platelet aggregation induced by ADP; and they increased the time required for artery occlusion without prolonging the bleeding time. Taken together these findings represent a significant contribution to the field and pave the way for the development of novel antiplatelet drugs. The approach taken in this study is logical, the experimental work sound and the conclusions drawn from the results highly relevant.

Specific points and suggested changes:

11. Page 2, line 57:  Are these Kiapp values contained in reference 10 by Salu et al., 2019?

22. The sequences of pDrTI and pDrTI-RI are not clear from the Methods and Results sections. It is only in he Discussion that the authors elaborate on the design of the peptides. It would be preferable to mention this in the Materials and Methods and indicate the sequences of the peptides.

33. Clearly, the target receptor of these proteins is of great interest and should be pursued in future studies. Have the authors done any preliminary experiments to see if they bind to target proteins, such as the B2 receptor?

44. The stability of these enantiomeric peptides is key to their efficacy. Have the authors done any work investigating the stability and half-life of the peptides.

55.  Page 2, Line 45-46 needs reworking as there is redundancy in the sentence

66. Page 2, Line 73: substrate should be plural

77. Figure 1: treatment symbols should be more distinguishable, particularly pDrTI and pDrTI-RI or the size of symbols made more visible

88. Page 7, Line 269-270 appears to have words or a section of the sentence missing. Is this meant to be a heading?

Author Response

 As respostas aos revisores estão anexadas.

Reviewer 2 Report

Figures 1,2, 3 and 7 lack the information regarding number of repetitions and on some of these figures no error bars are presented. This does not allow to evaluate whether experimental data support conclusions.

Why aspirin is presented as a control when it is not active in experimental conditions? It is not obligatory to present all experimental data obtained. Especially controls which don’t work.

Author Response

(The authors gave the same response as above.)

Round 2

Reviewer 2 Report

Figures 2,3 and some data in 7 still lack error bars and raw data points are not shown. Therefore it is impossible to have an impression on the variance. It is not shown in the figures which differences are significant. And yet the authors apparently know how to present scientific data as shown by Figure 4B. I see no excuse why other data could not be presented in exactly same way. Without it is impossible to evaluate if the conclusions are supported by the results.

Author Response

We modified the manuscript figure as requested.

Thank you for all your consideration 
